# A Hydrothermal Method to Generate Carbon Quantum Dots from Waste Bones and Their Detection of Laundry Powder

**DOI:** 10.3390/molecules27196479

**Published:** 2022-10-01

**Authors:** Heng Ye, Binbin Liu, Jin Wang, Chunyu Zhou, Zhili Xiong, Longshan Zhao

**Affiliations:** 1School of Pharmaceutical Engineering, Shenyang Pharmaceutical University, Shenyang 110016, China; 2Xiamen Cardiovascular Hospital, School of Medicine, Xiamen University, Jinshan Road 2999, Xiamen 361015, China; 3Interdisciplinary Science and Engineering in Health Systems, Institute of Academic and Research, Okayama University, Okayama 700-8530, Japan; 4School of Pharmacy, Shenyang Pharmaceutical University, 103 Wenhua Road Shenhe District, Shenyang 110016, China

**Keywords:** biomass, waste chicken bones, carbon quantum dots, CQDs, fluorescent probe, laundry powder

## Abstract

Surfactants are one of the major pollutants in laundry powder, which have an impact on the environment and human health. Carbon quantum dots (CQDs) are spherical zero-dimensional fluorescent nanoparticles with great potential for fluorescent probing, electrochemical biosensing and ion sensing. Herein, a bottom-up approach was developed for the synthesis of CQDs from biomass to detect laundry detergent and laundry powder. Waste chicken bones were used as carbon precursors after being dried, crushed and reacted with pure water at 180 °C for 4 h to generate CQDs, which exhibited a monodisperse quasi-spherical structure with an average particle size of 3.2 ± 0.2 nm. Functional groups, including -OH, C=O, C=C and C-O, were identified on the surface of the prepared CQDs. The optimal fluorescence excitation wavelength of the yellow-brown CQDs was 380 nm, with a corresponding emission peak at 465 nm. CQDs did not significantly increase cell death in multiple cell lines at concentrations of 200 µg·mL^−1^. Fluorescence enhancement of CQDs was observed after addition of sodium dodecyl benzene sulphonate, a major anionic surfactant in laundry powder. A linear relationship between fluorescence enhancement CQDs and the concentration of laundry powder was established. Thus, a hydrothermal method was developed to generate CQDs from waste biomass that may be used as a fluorescent probe to detect laundry powder.

## 1. Introduction

The consumption of laundry powder is showing a continuous upward trend with economic growth and the improvement of living standards. The main components of laundry powder are builders, including phosphates, surfactants, bleaches, enzymes and other ingredients [1]. Phosphates and surfactants are two major pollutants in laundry powder, which have an impact on the environment. Since laundry powder without phosphates has been widely used, residue of surfactants became an important environmental problem, which showed dose-dependent toxic effects on human bronchial epithelial cells [2] and multiple trophic levels including microalgae, cladocerans, ostracods, amphipods, macrophytes and fish [3]. Therefore, it is important to develop a method to trace the residue of laundry detergent and laundry powder.

Several methods were reported to detect laundry powder, including liquid chromatography tandem mass spectrometry to quantify enzymes in laundry powder [4], nuclear magnetic resonance spectroscopy to identify detergent formulations [5], infrared absorption spectra to determine the detergents [6] and two-phase titration method for determining cationic surfactant [7]. However, these methods need complex operation or expensive equipment. Thus, we would like to develop an easier method to trace the residue of laundry powder.

Carbon quantum dots (CQDs) are spherical zero-dimensional fluorescent nanoparticles with sizes below 10 nm [8]. They are promising nanomaterials for bioimaging [9], drug delivery [10], electrochemical biosensing [11] and ion sensing [12]. CQDs have received a great deal of attention because they exhibit high luminescence and good solubility and are thus also called carbon nano-lights. They are used as fluorescent probes to detect biomolecules [13] and chemical substances [14]. Functional groups including carboxyl and hydroxy groups on the surface of CQDs are associated with their biocompatibility and water solubility. Moreover, the size, shape, functional groups and surface doping of CQDs determine their optical properties, including the spectrum of light emission, tuneable fluorescence and efficiency of multiphoton up-conversion and down-conversion [15]. The CQDs from different sources have been reported to detect metal ion [16], 2,4,6-Trinitrotoluene [17], pyrophosphate [18] and so on. Our previous reports also indicated that CQDs can be used to identify acetylcholinesterase [19] and α-glucosidase activity as a fluorescent probe [20]. Therefore, we would like to develop a method to trace laundry powder by using the fluorescent nanoparticles CQDs.

The synthetic methods for CQDs can be classified into “top-down” and “bottom-up” strategies. Top-down synthesis starts from carbon nanotubes, carbon fibres, graphite rods, carbon ash and activated carbon. By contrast, bottom-up synthesis begins from carbon-containing compounds such as small organic molecules or oligomers [12,21]. Hydrothermal approaches [22,23], microwave methods [24] and thermal decomposition [25] are widely used to generate CQDs from various carbon precursors such as glucose [26] and lemon juice [27]. However, techniques to generate CQDs from large biomass waste remain limited.

CQDs have been generated from lemon peel waste [28] and dried leaves and flowers [29]. However, synthetic methods for CQDs from different sources vary, and the optical properties and usages of CQDs from different biomass sources are also different [30]. The animal bone is an easily accessible waste during meat processing activity. Slaughterhouses produce about 130 billion kg of animal bone residue globally every year [31]. In most developing countries, the waste bones are disposed of without regard to sound environmental management practices, making them harmful to humans and other terrestrial and aquatic life [32]. Thus, developing a method to generate CQDs from animal bone to detect laundry powder is a low-cost and environmentally friendly approach.

Herein, waste raw bones from chicken were used as carbon precursors to generate CQDs. The size, shape, functional groups, cytotoxicity, optical properties and stabilities of the prepared CQDs were measured. Finally, CQDs were found to exhibit increased fluorescence intensity after laundry detergent addition. Thus, we developed a method to generate CQDs, reuse waste biomass and detect laundry powder in a low-cost and environmentally friendly manner.

## 2. Results

### 2.1. Functional Groups of the Prepared CQDs

CQDs were prepared from waste bones as described in the methods (Figure 1). Transmission electron microscopy (TEM) images of CQDs are shown in Figure 2a. CQDs displayed good dispersion and uniform particle size averaging 3.2 ± 0.2 nm (Figure 2b). 

The X-ray diffractometer (XRD) profile of CQDs (Figure 3a) indicated that there was a broad (002) peak at 21.15°, which corresponds to the graphite structure [33,34]. Fourier transform infrared spectrometry (FT-IR) analysis of CQDs (Figure 3b) revealed (1) strong and broad characteristic peaks at 3150–3500 cm^−1^ assigned to the stretching vibration of the -OH group [35,36]; (2) an asymmetric vibration at 2644 cm^−1^ corresponding to -CH_3_ [37,38]; (3) a characteristic infrared peak at 1620 cm^−1^ assigned to the stretching vibration of C=O; (4) a characteristic peak at 1385 cm^−1^, which is the out-of-plane bending vibration of the olefin C-H.

Full-scan X-ray photoelectron spectrometry (XPS) analysis of CQDs (Figure 3c) indicated two characteristic peaks at 285.7 eV and 530.9 eV, which belong to C1s and O1s. The high-resolution XPS spectrum of C1s indicates that peaks at 283.85 eV, 284.87 eV and 287.70 eV represent C=C/CC, C-O C=O, respectively (Figure 3d). Characteristic peaks of -OH at 531.1 eV, 531.9 eV and 532.5 eV were observed in the high resolution XPS spectrum of O1s (Figure 3e) [39]. In addition, the relative amounts of C and O in CQDs were 75.34% and 19.49%, respectively. These results indicated that the surface of CQDs had functional groups such as -OH, C=O, C=C and C-O, which were conducive to enhancing solubility in water.

### 2.2. Optical Properties of CQDs

CQDs in aqueous solution were yellow-brown and transparent, and they emitted light blue fluorescence under 365 nm UV light irradiation (Figure 4a). Photoluminescence (PL) spectra of CQDs was measured at excitation wavelengths from 360 to 405 nm (Figure 4b). The emission wavelength was shifted when the excitation wavelength was increased. The strongest fluorescence intensity of CQDs was observed when the excitation wavelength was 380 nm, with the corresponding emission peak at 465 nm. The results suggested that CQDs had an excitation-dependent PL feature, and an excitation wavelength of 380 nm was therefore selected as the optimal excitation wavelength in this work.

### 2.3. Optical Stabilities of CQDs under Different pH, Salt Concentration, Light and Temperature Conditions

The highest fluorescence intensity of CQDs was observed when the pH of the solution was 7.21 (pure water). The fluorescence intensity of CQDs was slightly decreased when the pH was 2, 4, 6 or 8 compared to 7.21. The fluorescence intensity of CQDs was relatively stable when the pH of CQD solutions was between 2 and 6. When the pH was >8, the fluorescence intensity was sharply decreased (Figure 5a).

The fluorescence intensity of CQDs was reduced when the concentration of NaCl was increased, and the fluorescence intensity was relatively stable when the concentration of NaCl was increased from 0.25 to 2 mol·L^−1^. The highest fluorescence intensity of CQDs was observed in salt-free conditions (Figure 5b).

There was a slight decrease in the fluorescence intensity of CQDs when the temperature of the solution was increased from 15 to 45 °C. A sharp decrease in fluorescence intensity of CQDs was observed when the temperature was >45 °C (Figure 5c). The results indicate that the fluorescence intensity of CQDs is relatively stable when the temperature of the solution is between 15 °C and 45 °C. 

To explore the photostability of CQDs, the fluorescence intensity of CQD solutions was measured after 5, 10, 15, 20, 25 and 30 min of continuous Xe lamp irradiation. The fluorescence intensity of CQDs did not decrease after 30 min of Xe lamp irradiation (Figure 5d), indicating that CQDs have strong resistance to bleaching. 

### 2.4. CQDs Exert Low Cytotoxicity

The cytotoxicity of CQDs was studied in multiple cell lines, including hepatocellular carcinoma, Hepa 1-6, HCT-116 colon cancer cells and primary mouse embryonic fibroblasts (MEFs). The cell survival rates after treatment with 1.25 to 20 µg·mL^−1^ CQDs did not significantly decrease (*p* > 0.05) compared with non-treatment controls (Figure 6a), and cell morphology did not change when cells were treated with 20 µg·mL^−1^ CQDs for 48 h (Figure 6b). The results indicated that CQDs did not exhibit cytotoxicity when the concentration was increased to 20 µg·mL^−1^, implying good biocompatibility.

### 2.5. Laundry Detergents Enhance the Fluorescence Intensity of CQDs

Flow cytometry was used to analyse the relative fluorescence intensity of the prepared CQDs. Forward scatter (FSC) represents the size of the prepared CQDs (Figure 7a). Approximately 80% of prepared CQDs were located in a narrowed FSC region. The relative fluorescence intensity of prepared CQDs in the G1 region was quantified in solutions with or without the anionic surfactant sodium dodecyl benzene sulphonate (SDBS). The fluorescence intensity of CQDs was increased in solutions with SDBS compared to those without (Figure 7b). Because SDBS is a major component of laundry powder, Liby, a widely used laundry powder, was used to explore whether the CQDs could be used to detect the residual laundry powder in water samples. Increased fluorescence intensity of CQDs was observed after addition of Liby (Figure 7b). Thus, the results indicated that the prepared CQDs could be used to detect residual laundry powder.

Further, we measured the fluorescence intensity of prepared CQDs after addition of an anionic surfactant sodium dodecyl sulfate (SDS), or a nonionic surfactant NP40, which are absent in laundry powder. The fluorescence intensity of CQDs did not increase in solutions with SDS or NP40 compared to those without (Figure 7b). Moreover, the fluorescence intensity of CQDs did not increase in solutions with a lentivirus compared to those without (Figure 7b). Further, the fluorescence intensity of CQDs was quantified in solutions containing 10 mmol·L^−1^ Mg^2+^, Na^+^, K^+^, Ca^2+^, Zn^+^, Al^3+^, Cu^2+^ or vitamin C (Table 1). There was no fluorescence intensity enhancement in these solutions. Consistently, the fluorescence intensity of CQDs was increased from 1968 ± 2.31 to 3207 ± 10.0 after Liby addition (Figure 7c). Thus, the prepared CQDs could detect laundry powder and laundry detergent. 

### 2.6. CQDs Can Be Used as a Fluorescence Probe to Detect Laundry Powder

First, the fluorescence stability of the reaction system between CQDs and Liby was analysed under different conditions. When the pH of the reaction system was between 2 and 10, Liby-induced fluorescence enhancement of CQDs was relatively stable (Figure 8a). Additionally, Liby-induced fluorescence enhancement of CQDs decreased with increasing reaction time (Figure 8b). Therefore, the reaction time between CQDs and Liby was set at 3 min, and the fluorescence intensity was measured within 5 min. 

Second, PL spectra of prepared CQDs were measured in solutions with or without different concentrations of Liby. Liby-induced fluorescence enhancement of CQDs was dose-dependent (Figure 8c). The fluorescence intensity (F_0_) of CQD solutions and the fluorescence intensity (F) of CQD solutions with different concentrations (C) of Liby was measured. The results’ linear regression equation (r^2^ = 0.9925) was (F − F_0_)/F_0_ = 0.0327C − 0.0014, and the limit of detection for Liby was 0.17 μg·mL^−1^ (Figure 8d).

Finally, the recovery rates for Liby in water samples from three sources were measured to evaluate the validity of the assay using prepared CQDs. The recovery rates of Liby in the three water samples were 87.2–104.2%, 85.2–94% and 95.3–101.6%, respectively (Table 2). The limit of quantification was defined as the percent relative standard deviation (% RSD). The low concentration point RSD was <5.3% (*n* = 3). This indicated that CQDs could be used to detect laundry powder in water samples from different sources. 

## 3. Discussion and Conclusions

It has been widely reported that CQDs can be produced from various kinds of biomass, including plants and their derivatives such as corn powder [40] and green tea [41]; animals and their derivatives such as crab shell [42] and honey [43]; and municipal waste such as coal [44]. Herein, a one-step hydrothermal synthesis method was developed to generate CQDs using waste bone from chicken as raw material. Hydrothermal synthesis is a chemical reaction method in aqueous solution under the conditions of temperature of 100–1000 °C, and the product has high purity, good dispersion and easy particle size control [45]. The size of CQDs generated from crab shell by hydrothermal method is 4.0 ± 0.7 nm [42], and the average size of CQDs from lemon juice by hydrothermal method is 3.1 nm [27]. Consistently, the prepared CQDs had good water solubility and an almost monodisperse quasi-spherical structure with an average particle size of 3.2 ± 0.2 nm, which conforms to the ideals of green chemistry. 

The optimal fluorescence excitation wavelength of the CQDs was 380 nm, and the corresponding emission peak was 465 nm. Therefore, emission of blue-green light was visible by the naked eyes under 380 nm UV light irradiation. The optical properties of prepared CQDs are quite similar with some published literature [27,42,45,46]. For example, CQDs synthesized from organic waste exhibits a 450 nm blue fluorescent light [47]. Besides the blue light with emission wavelengths of 450–475, the emission wavelengths of CQDs were observed in the range of 380–450, 476–495, 495–570, 570–590, 590–620 and 620–750 nm [45]. However, the PL mechanism of CQDs from biomass remains largely unknown. CQDs from the 3,4,9,10-perylenetetracarboxylic dianhydride with the sizes of 3.01 ± 0.32 and 4.32 ± 0.38 exhibit 400 and 480 nm fluorescence. Their PL mechanism is associated with the functional groups including –COOH, –NH2 and C–N on the surface, which induce fluorescence via the defect-induced luminescence mechanism of conjugated π structure [48]. Herein, the functional groups of prepared CQDs contain -OH, C=O, C=C and C-O, which correspond to the π–π* and n–π* transition [49]. Increased fluorescence intensity of the CQDs was observed after addition of SDBS but not SDS and NP40, which have quite similar structures as SDBS. Compared to SDS and NP40, SDBS contains an ionic conjugated system (Figure 7b). Whether SDBS-induced fluorescence enhancement of CQDs is associated with the interaction between ionic conjugated system in SDBS and the π–π* and n–π* transition of CQDs needs further analysis. Thus, this conjugated structure in CQDs is probably associated with its optical properties and PL mechanism. 

Considering that laundry powder contains more than 20 components [1] (main component SDBS and other ingredients) and other interferences in water samples, the influence on the fluorescence intensity was investigated. The results demonstrated that SDBS and laundry powder could significantly increase the fluorescence intensity of the CQDs, whereas other ingredients and water pollutants or interferences such as SDS, virus and NP40 did not showed increased fluorescence intensity. Furthermore, real water samples were applied to testify the CQDs probe in detection of laundry powder with recovery rates more than 85%, which suggested that the prepared CQDs could successfully detect laundry powder in some real water samples. 

CQDs from biomass were widely used to detect metal ions [50,51]. The CQDs from pear juice could selectively detect Fe^3+^ [52]. On the other hand, non-metal ionic molecules, such as Sudan I [53] and promethazine hydrochloride [54], can be detected by the CQDs from different biomass. Our work herein reported that CQDs such prepared could detect laundry powder and SDBS. Moreover, CQDs were reported to improve the photocatalysis to remove environmental pollutants [55]. Nitrogen–sulfur codoped carbon quantum dots were reported to detect sulfide and ferric ion, which may be used to detect inorganic pollutants in water samples [56]. CQDs that have been combined with TiO_2_ were reported to disinfect wastewater [57]. Thus, CQDs are promising nanomaterials for water pollution treatment [15].

In summary, a synthesis method for CQDs was developed using waste bones from chicken. The prepared CQDs possess unique optical properties that may allow them to be used as a fluorescent probe to detect laundry detergents. This method could assist the low-cost and environmentally friendly reuse of waste biomass.

## 4. Materials and Methods

### 4.1. Instruments and Materials

A UV-5500 PC UV-Spectrophotometer (Metash Instruments, Shanghai, China), an F97 Fluorescence Spectrophotometer (Cold Light Technology Co., Shanghai, China), a JEM- 2100 transmission electron microscope (JEOL, Tokyo, Japan), a D8 Advance X-ray diffractometer (Bruker, Karlsruhe, Germany), a Nicolet IS50 Fourier transform infrared spectrometer (Thermo Scientific, Waltham, MA, USA), an ESCALAB 250Xi X-ray photoelectron spectrometer (Thermo Scientific), a PHS 3W pH meter (Lei Magnetic Co., Shanghai, China), a Varioskan LUX microplate reader (Thermo Scientific), an EVOS FL Auto 2 microscope (Thermo Scientific), and an LSR Fortessa Flow cytometer (BD, Franklin Lakes, NJ, USA) were used in this research.

Reagents and materials were from the following sources: pure water (Wahaha, Hangzhou, China), HCl (analytical-grade; RionLon Chemical Co., Tianjin, China), vitamin C, CaCl_2_, FeCl_3_, MgCl_2_, ZnCl_2_, KCl, AlCl_3_, CuCl_2_ and NaCl (analytical-grade; Berens Biochemical Co., Ltd., Tianjin, China), Dulbecco’s Modified Eagles Medium (DMEM; high-glucose; Procell, Wuhan, China), a Cell Counting Kit-8 (CCK-8; Cellcook, Guangzhou, China), fetal bovine serum (FBS; Gibco, Waltham, MA, USA), sodium dodecyl benzene sulphonate (analytical-grade; D&B Biological Science and Technology Co., Shanghai, China), Liby laundry powder (main component SDBS; Liby, Guangzhou, China), Sodium dodecyl sulfate (analytical-grade, Solarbio Science & Technology Co., Beijing, China), NP40 (Solarbio Science & Technology Co., Beijing, China).

### 4.2. CQDs Synthesis

Waste raw bones from chicken were obtained from the cafeteria of Shenyang Pharmaceutical University, washed, dried in a drying oven at 60 °C for 2 h, and crushed. A 1 g sample of crushed chicken skeleton powder was reacted with 50 mL pure water in a 100 mL reaction kettle. The reaction was performed in the oven at 180 °C for 4 h (Figure 1). After cooling, CQD solution was obtained from the supernatant after centrifugation for 10 min. CQD powder was prepared by freeze-drying, and 0.1 mg·mL^−1^ CQD solutions were prepared using pure water.

### 4.3. Fluorescence Intensity Measurement by Fluorescence Spectrophotometer

A 100 μL sample of CQD solution (0.1 mg·mL^−1^) was diluted to 3 mL for fluorescence intensity, excitation and emission spectra measurement by a fluorescence spectrophotometer. Liby, NaCl, Mg^2+^, Na^+^, K^+^, Ca^2+^, Zn^+^, Al^3+^, Cu^2+^ or vitamin C was added to the solution at the indicated concentration. All experiments were performed in triplicate.

### 4.4. Lentivirus Preparation

HEK 293T cells were transfected with packaging vectors and an empty plasmid. The lentivirus was collected from the supernatant after 48 h. The concentration of lentivirus was quantified.

### 4.5. Flow Cytometry

A 100 μL sample of CQD solution (0.1 mg·mL^−1^) was diluted to 1 mL with or without 2.5 μg·mL^−1^ SDBS or Liby. The relative fluorescence intensity of CQDs was measured by flow cytometry with a 405 nm laser and a 450/40 filter.

### 4.6. Cell Culture

Hepa 1-6 hepatocellular carcinoma cells, HCT-116 colon cancer cells and HEK 293T cells were obtained from the American Type Culture Collection (ATCC), and primary mouse embryonic fibroblasts (MEFs) were purified from mouse embryos at E14.5. A green fluorescent protein (GFP)-encoding vector was transfected into HCT-116 cells and stable expression of GFP was established. All cells were maintained in DMEM high glucose medium with 10% FBS and 100 U/mL penicillin and streptomycin at 37 °C under 5% CO_2_.

### 4.7. Cytotoxicity Test

Cells were seeded in 96-well plates and cultured in an incubator for 24 h. Cells were treated with 0, 1.25, 2.5, 5, 10 and 20 μg·mL^−1^ CQD solutions for 48 h. The cell survival rate was measured by CCK-8 assay, as described previously [58]. The optical density (OD) was measured by a microplate reader at 450 nm.

### 4.8. Statistical Analysis

For cytotoxicity tests, data were expressed as mean ± standard deviation (SD), and the difference between control and CQDs-treated groups was analysed using an unpaired two-tailed Student’s *t*-test (*p* < 0.05 was considered significant). Each group included three biological replicates.

## Figures and Tables

**Figure 1 molecules-27-06479-f001:**
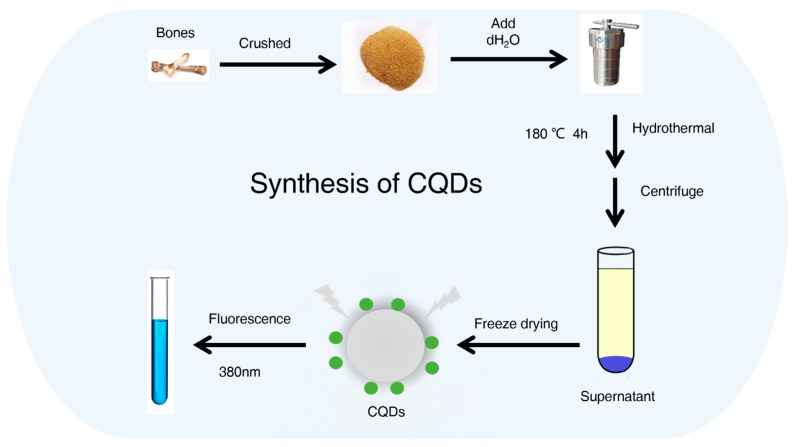
Procedure for preparing carbon quantum dots (CQDs) from waste chicken bones.

**Figure 2 molecules-27-06479-f002:**
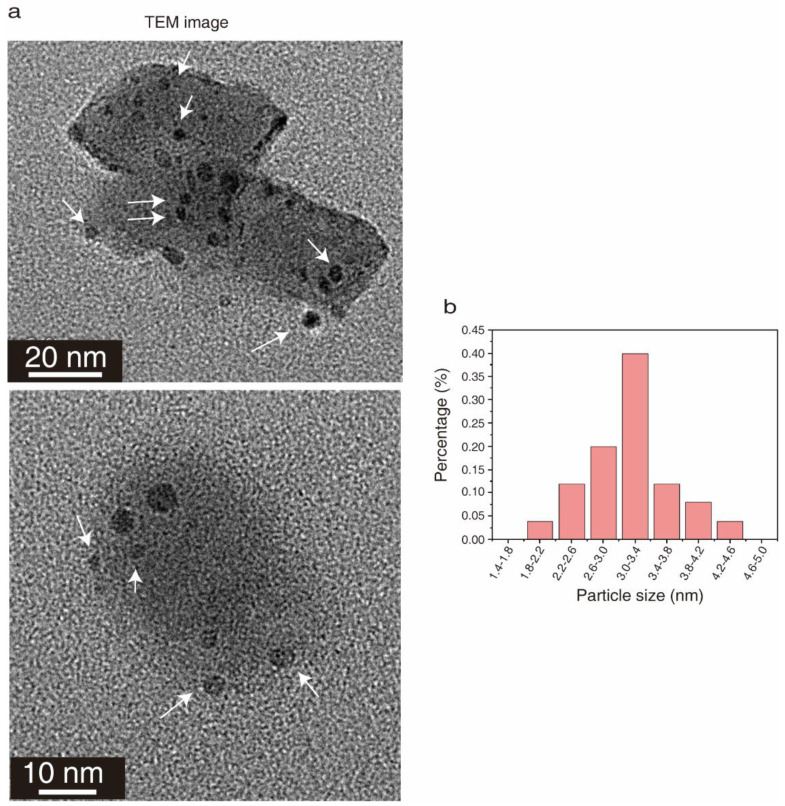
The shape and size of CQDs prepared from waste chicken bones. (**a**) TEM image of CQDs. The arrows indicated the prepared CQDs. (**b**) Size distribution of CQDs.

**Figure 3 molecules-27-06479-f003:**
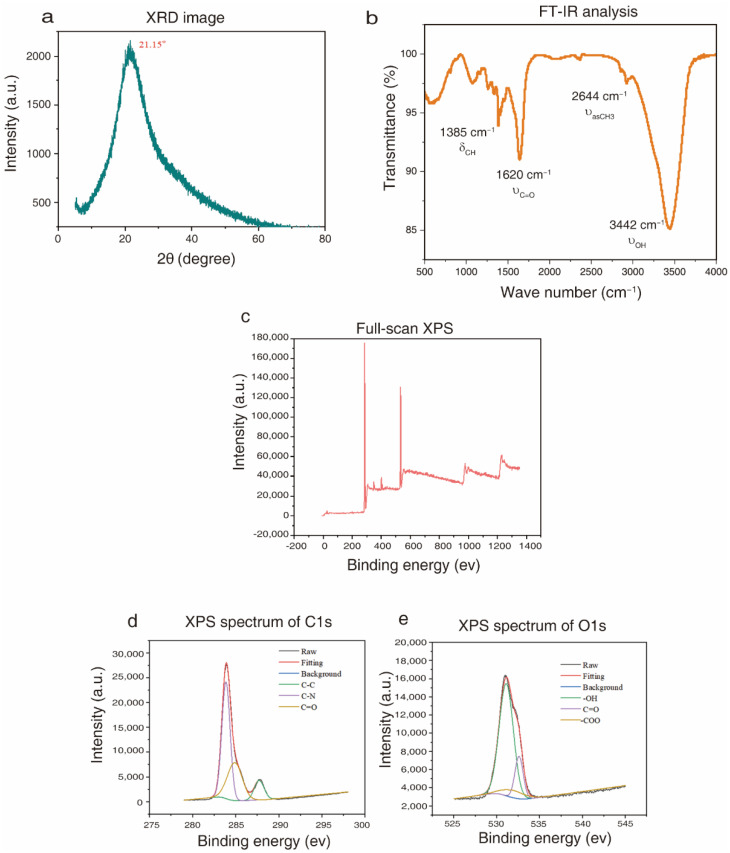
Functional groups of CQDs prepared from chicken bones. (**a**) XRD image of CQDs. (**b**) FT-IR analysis of CQDs. (**c**) Full-scan XPS spectrum of CQDs. (**d**) High-resolution XPS spectrum of C1s. (**e**) High-resolution XPS spectrum of O1s.

**Figure 4 molecules-27-06479-f004:**
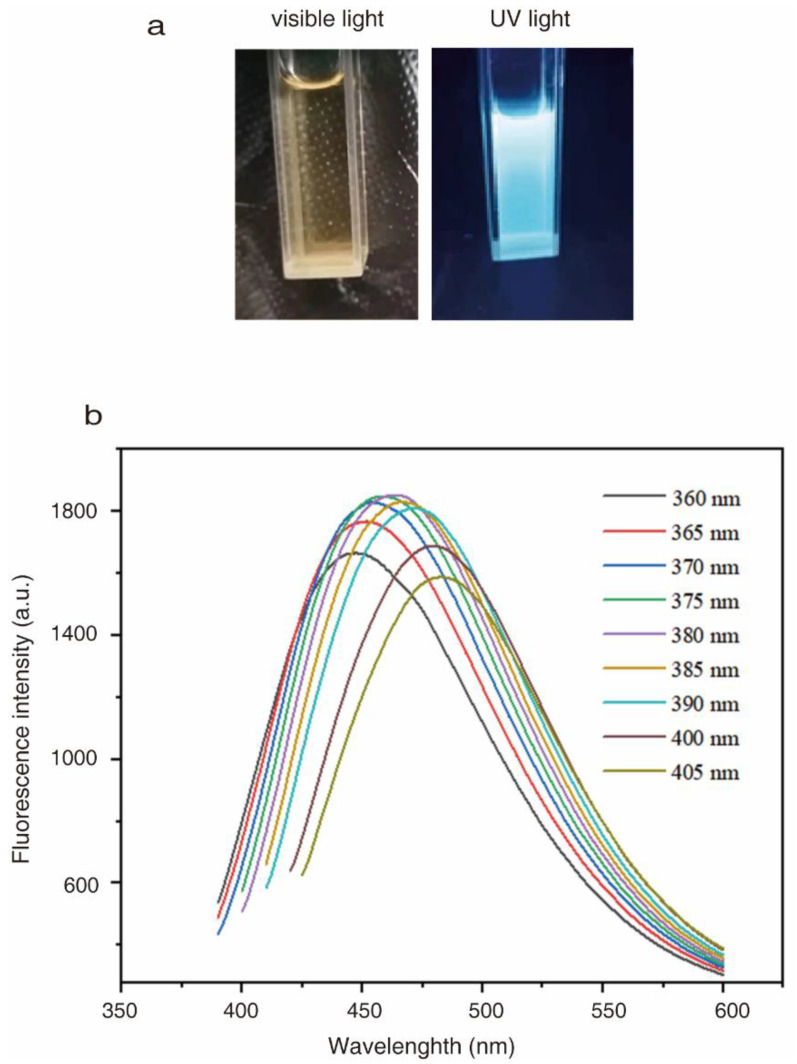
Optical properties of CQDs prepared from chicken bones. (**a**) Typical images of CQDs under UV and visible light irradiation are shown. (**b**) Excitation-dependent fluorescence spectra of CQDs.

**Figure 5 molecules-27-06479-f005:**
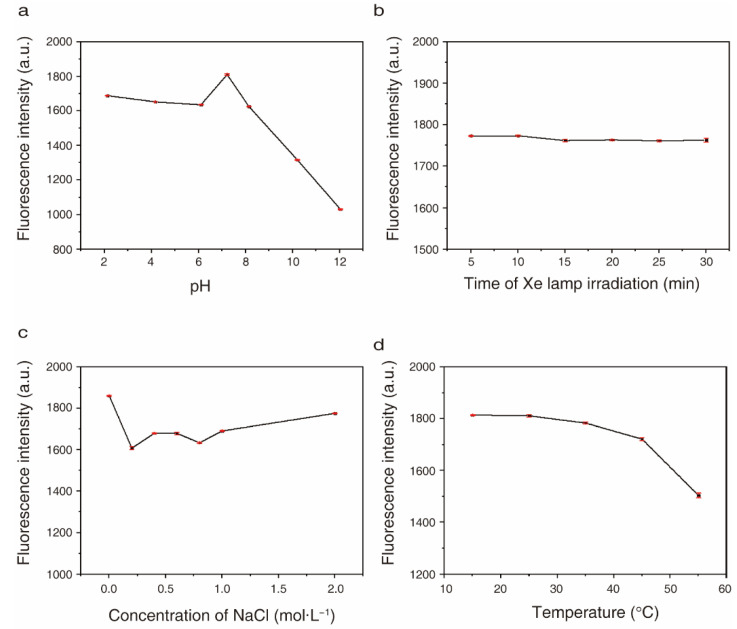
Fluorescence stability of CQDs prepared from chicken bones. The fluorescence intensity of CQD solutions was measured under the indicated (**a**) pH, (**b**) NaCl concentration, (**c**) temperature and (**d**) Xe lamp irradiation (5–30 min) conditions.

**Figure 6 molecules-27-06479-f006:**
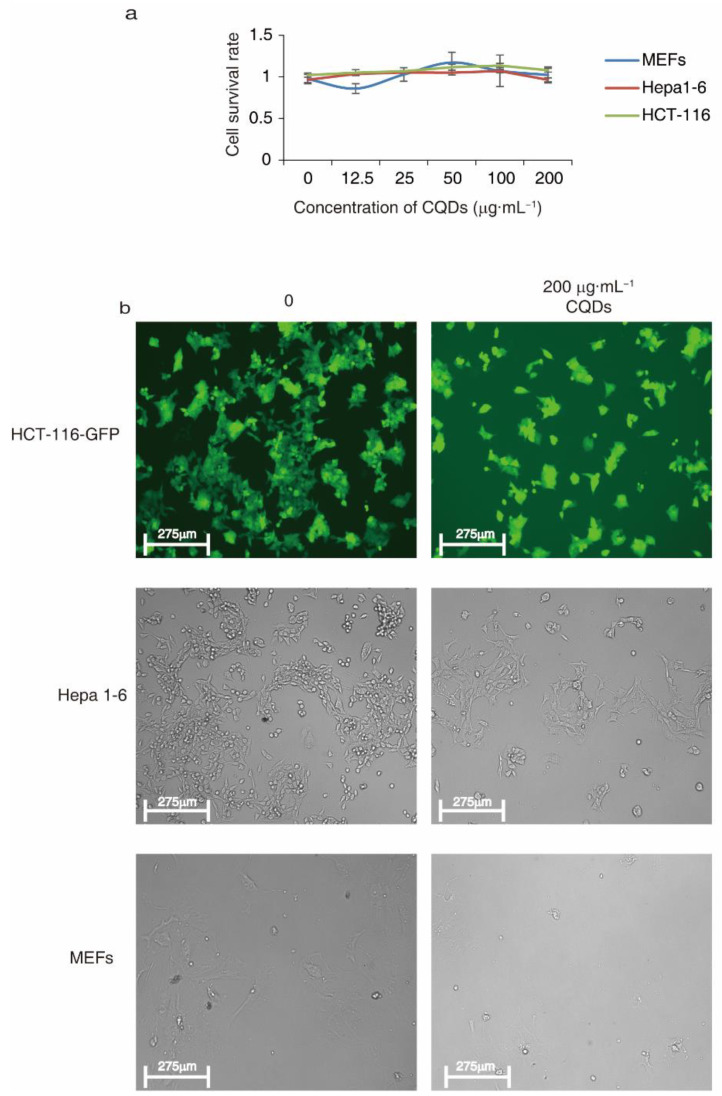
Cytotoxicity of CQDs prepared from chicken bones. (**a**) Cell survival rates measured in indicated cells by CCK-8 assay. The indicated concentrations of CQDs were used to treat cells for 48 h. (**b**) Cell morphology of HCT-116-GFP, Hepa 1-6 and MEFs after treatment with or without 200 μg·mL^−1^ CQDs for 48 h.

**Figure 7 molecules-27-06479-f007:**
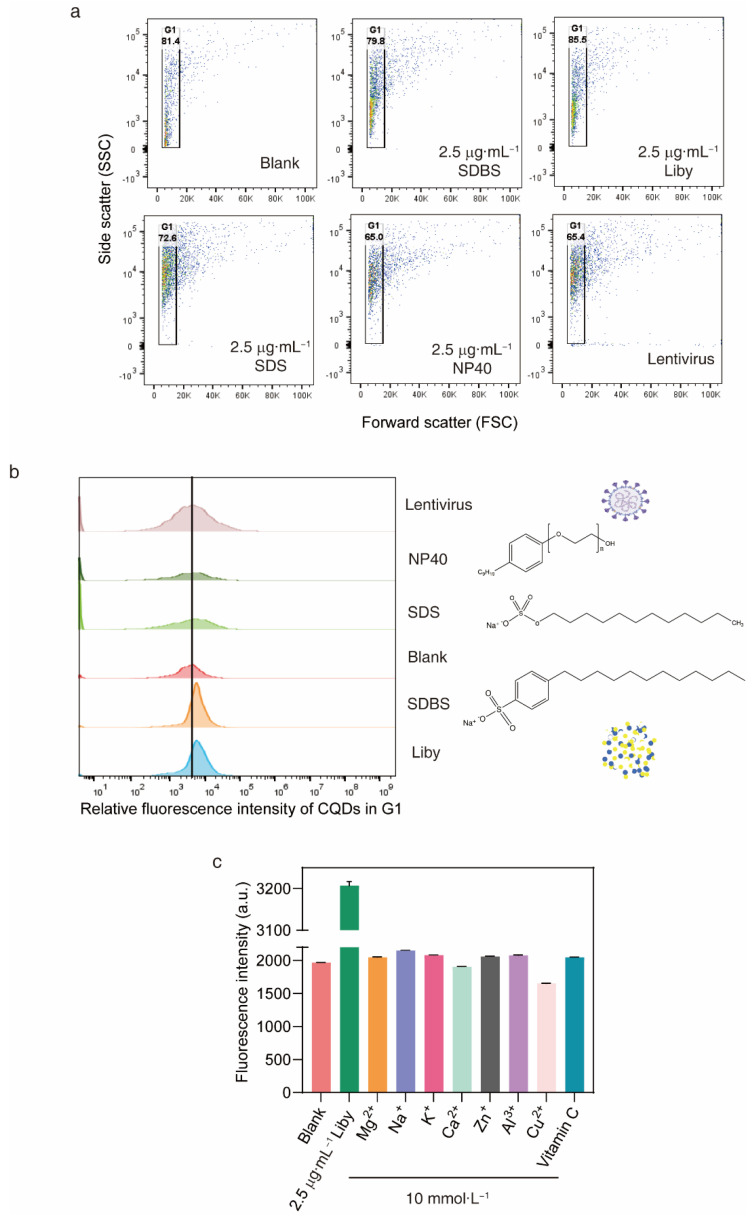
Laundry detergents enhance the fluorescence intensity of CQDs. (**a**) CQDs were assessed using FSC/SSC parameters. (**b**) Relative fluorescence intensity of CQDs in the G1 region quantified by flow cytometry with or without 2.5 μg·mL^−1^ SDBS, Liby, SDS, NP40 or 6.25 × 10^5^ lentiviral particles. (**c**) Fluorescence intensity of CQD solutions measured in the presence of 10 mmol·L^−1^ Mg^2+^, Na^+^, K^+^, Ca^2+^, Zn^+^, Al^3+^, Cu^2+^, vitamin C and 2.5 μg·mL^−1^ Liby.

**Figure 8 molecules-27-06479-f008:**
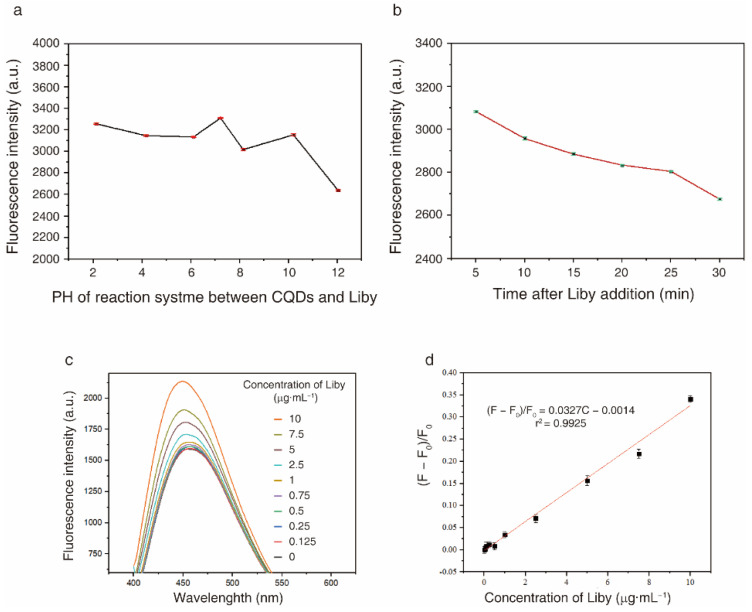
Fluorescence intensity of CQDs and 2.5 μg·mL^−1^ Liby solution measured (**a**) under different pH conditions and (**b**) at the indicated timepoints after Liby addition. (**c**) PL spectra of CQDs and Liby solution. (**d**) Fluorescence intensity (F_0_) of CQD solutions and the fluorescence intensity (F) of CQD solutions containing different concentrations (C) of Liby. A linear regression equation was obtained. All experiments were repeated three times. Results are expressed as mean ± standard deviation (SD).

**Table 1 molecules-27-06479-t001:** Fluorescence enhancement of CQDs under different treatments.

Component	MFI ± SD
blank	1968.33 ± 2.31
Liby	3207 ± 10.0
Mg^2+^	2050 ± 4.7
Na^+^	2153 ± 3.0
K^+^	2080 ± 4.6
Ca^2+^	1905 ± 4.0
Zn^+^	2062 ± 5.0
Al^3+^	2079 ± 5.2
Cu^2+^	1651.67 ± 2.08
Vitamin c	2048.33 ± 3.05

**Table 2 molecules-27-06479-t002:** Standard recovery of laundry detergents from different water samples (*n* = 3).

Samples	Added (μg·mL^−1^)	Found (μg·mL^−1^)	Recovery Rate (%)	RSD (%)
Mineral water	0.50	0.53	106.00	5.3
	2.50	2.22	88.80	3.3
	7.50	6.54	87.20	1.7
Tap water	0.50	0.47	94.00	4.49
	2.50	2.46	98.40	1.3
	7.50	6.39	85.20	1.23
River water	0.50	0.48	96.00	2.15
	2.50	2.54	101.60	1.01
	7.50	7.16	95.47	3.45

## Data Availability

Datasets are from the corresponding author on reasonable request. Data is available on request.

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
