# Peer review of "A Hydrothermal Method to Generate Carbon Quantum Dots from Waste Bones and Their Detection of Laundry Powder"

_molecules, 2022, doi:10.3390/molecules27196479_

Round 1

Reviewer 1 Report

The manuscript entitled “A hydrothermal method to generate carbon quantum dots from waste bones and their selective detection of laundry powder” aims to describe a bottom-up approach for the synthesis of CQDs from biomass and the application of synthesized CQDs for sodium dodecyl benzene sulphonate detection. The idea is good, but the manuscript is prepared poorly. Here are my concerns:

To equalize sodium dodecyl benzene sulphonate with laundry powder is a mistake.

What makes and proves this method selective? I am sure that some other water pollutants would also enhance CQDs' fluorescence intensity. It should be thoroughly discussed.

The Introduction is too short and not informative enough.

The results are well presented.

On the other hand, the discussion and conclusion are basically non-existing. It is a major problem. The authors should thoroughly discuss their results, especially with respect to up-to-date literature.

Author Response

The manuscript entitled “A hydrothermal method to generate carbon quantum dots from waste bones and their selective detection of laundry powder” aims to describe a bottom-up approach for the synthesis of CQDs from biomass and the application of synthesized CQDs for sodium dodecyl benzene sulphonate detection. The idea is good, but the manuscript is prepared poorly. Here are my concerns:

To equalize sodium dodecyl benzene sulphonate with laundry powder is a mistake.

Thank you for the comments. We add the new data in Figure 7b and add the following information.

In results section at lin194-205

To understand the selective detection of laundry detergent by CQDs, we measured the fluorescence intensity of prepared CQDs after addition of an anionic surfactant sodium dodecyl sulfate (SDS), or a nonionic surfactant NP40, which are absent in laundry powder. The fluorescence intensity of CQDs did not increase in solutions with SDS or NP40 compared to those without (Fig. 7b). Besides, the fluorescence intensity of CQDs did not increase in solutions with a lentivirus d compared to those without (Fig. 7b).

In Discussion and Conclusion at line 250-253

Moreover, laundry powder usually contains more than 20 components [1], and the residue of laundry powder may contain not only SDBS but also other ingredients. Thus we develop a quantified method to trace laundry powder by using a commercial laundry powder but not SDBS. 

What makes and proves this method selective? I am sure that some other water pollutants would also enhance CQDs' fluorescence intensity. It should be thoroughly discussed.

Thank you for the questions. We add the following new data in Figure 7b and add the following information.

To understand the selective detection of laundry detergent by CQDs, we measured the fluorescence intensity of prepared CQDs after addition of an anionic surfactant sodium dodecyl sulfate (SDS), or a nonionic surfactant NP40, which are absent in laundry powder. The fluorescence intensity of CQDs did not increase in solutions with SDS or NP40 compared to those without (Fig. 7b). Besides, the fluorescence intensity of CQDs did not increase in solutions with a lentivirus d compared to those without (Fig. 7b). Further, the fluorescence intensity of CQDs was quantified in solutions containing 10 mmol·L-1 Mg2+, Na+, K+, Ca2+, Zn+, Al3+, Cu2+ or vitamin C (Table I). There was no fluorescence intensity enhancement in these solutions. Consistently, the fluorescence intensity of CQDs was increased from 1968 ± 2.31 to 3207 ± 10.0 after Liby addition (Fig. 7c). Thus, the prepared CQDs could selectively detect the laundry powder and laundry detergent.

In Discussion and Conclusion at line 244-263

Increased fluorescence intensity of the CQDs was observed after addition of SDBS but not SDS and NP40, which have quite similar structure as SDBS. Compared to SDS and NP40, SDBS contains an ionic conjugated system (Fig. 7b). The functional groups of such prepared CQDs contain -OH, C=O, C=C and C-O, which correspond to the π–π* and n–π* transition [40]. Whether SDBS-induced fluorescence enhancement of CQDs is associated the interaction between ionic conjugated system in SDBS and the π–π* and n–π* transition of CQDs needs further analysis. Moreover, laundry powder usually contains more than 20 components [1], and the residue of laundry powder may contain not only SDBS but also other ingredients. Thus we develop a quantified method to trace laundry powder by using a commercial laundry powder but not SDBS.  

A linear relationship between CQDs and laundry powder was observed. The enhanced fluorescence intensity of CQDs induced by laundry powder could be used to trace residues of laundry powder in water samples. The water pollutants in river water contains bacteria, viruses, parasites, fertilisers, pesticides, nitrates, phosphates, plastics, faecal waste and others [41]. Virus did not induce increased fluorescence intensity of the CQDs. And residue of laundry powder can be detected in river water, mineral water and tap water with the recovery rates more than 85%. This suggested that the prepared CQDs could be selectively detect the laundry powder in some real waters. However, whether high doses of other water pollutants affected on this assay need further analyze. Therefore, the prepared CQDs could be used as a fluorescent probe to detect residue of laundry powder in tap water, mineral water and tap water.

The Introduction is too short and not informative enough.

Thank you. We modified this part.

  1. Introduction

The consumptions of laundry powder are showing a continuous upward trend with economic growth and the improvement of living standards. The main components of laundry powder are builders including the phosphates, surfactants, bleaches, enzymes and other ingredients [1]. Phosphates and surfactants are two major pollutants in laundry powder which are impact on the environments. Since the laundry powder without phosphates has been widely used, residue of surfactants became an important environmental problem which showed dose-dependent toxic effects on human bronchial epithelial cells [2] and multiple trophic levels including microalgae, cladocerans, ostracods, amphipods, macrophytes and fish [3]. Therefore, it is important to develop a method to trace the residue of laundry detergent and laundry powder.

Several methods were reported to detect the laundry powder including liquid chromatography tandem mass spectrometry to quantify enzymes in laundry powder [4], nuclear magnetic resonance spectroscopy to identify detergent formulations [5], infrared absorption spectra to determine the detergents [6] and two-phase titration method for determining cationic surfactant [7]. However, these methods need complex operation or expensive equipment. Thus, we would like to develop an easier method to trace the residue of laundry powder.

Carbon quantum dots (CQDs) are spherical zero-dimensional fluorescent nanoparticles with sizes below 10 nm [8]. They are promising nanomaterials for bioimaging [9], drug delivery [10], electrochemical biosensing [11] and ion sensing [12]. CQDs have received a great deal of attention because they exhibit high luminescence and good solubility, and are thus also called carbon nano-lights. They are used as fluorescent probes to detect biomolecules [13] and chemical substances [14]. Functional groups including carboxyl and hydroxy groups on the surface of CQDs are associated with their biocompatibility and water solubility. Moreover, the size, shape, functional groups and surface doping of CQDs determine their optical properties, including the spectrum of light emission, tunable fluorescence, and efficiency of multiphoton up-conversion and down-conversion [15]. The CQDs from different sources have been reported to detect metal ion [16], 2,4,6-Trinitrotoluene [17] , pyrophosphate [18] and so on. Our previous reports also indicated that the CQDs can be used to identify acetylcholinesterase [19] and α-glucosidase activity as a fluorescent probe [20]. Therefore, we would like to develop a method to trace the laundry powder by using the fluorescent nanoparticles CQDs.

The synthetic methods for CQDs can be classified into “top-down“ and “bottom-up“ strategies. Top-down synthesis starts from carbon nanotubes, carbon fibers, graphite rods, carbon ash and activated carbon. By contrast, bottom-up synthesis begins from carbon-containing compounds such as small organic molecules or oligomers [12,21]. Hydrothermal approaches [22,23], microwave methods [24] and thermal decomposition [25] are widely used to generate CQDs from various carbon precursors such as glucose [26] and lemon juice [27]. However, techniques to generate CQDs from large biomass waste remain limited.

CQDs have been generated from lemon peel waste [28] and dried leaves and flowers [29]. However, synthetic methods for CQDs from different sources vary, and the optical properties and usages of CQDs from different biomass sources are also different [30].  The animal bone is an easily accessible waste during the meat processing activity. Slaughterhouses produced about 130 billion kg of animal bone residue globally every year [31]. In most developing countries, the waste bones are disposed of without regard to sound environmental management practices, making them harmful to humans and other terrestrial and aquatic life [32]. Thus, developing a method to generate CQDs from animal bone to detect the laundry powder is low-cost and environmentally friendly approach.

Herein, waste raw bones from chicken were used as carbon precursors to generate CQDs. The size, shape, functional groups, cytotoxicity, optical properties and stabilities of the prepared CQDs were measured. Finally, CQDs were found to exhibit increased fluorescence intensity after laundry detergent addition. Thus, we developed a method to generate CQDs, reuse waste biomass and detect the laundry powder in a low cost and environmentally friendly manner.

The results are well presented.

On the other hand, the discussion and conclusion are basically non-existing. It is a major problem. The authors should thoroughly discuss their results, especially with respect to up-to-date literature.

Thank you. We modified this part.

  1. Discussion and Conclusion

Herein, a one-step hydrothermal synthesis method was developed to generate CQDs using waste bone from chicken as raw material. The prepared CQDs had good water solubility and low toxicity, which conforms to the ideals of green chemistry.

The prepared CQDs had an almost monodisperse quasispherical structure with an average particle size of 3.2 ± 0.2 nm. The optimal fluorescence excitation wavelength of the CQDs was 380 nm, and the corresponding emission peak was 465 nm. Therefore, emission of blue-green light was visible by the naked eyes under 380 nm light irradiation.

Increased fluorescence intensity of the CQDs was observed after addition of SDBS but not SDS and NP40, which have quite similar structure as SDBS. Compared to SDS and NP40, SDBS contains an ionic conjugated system (Fig. 7b). The functional groups of such prepared CQDs contain -OH, C=O, C=C and C-O, which correspond to the π–π* and n–π* transition [40]. Whether SDBS-induced fluorescence enhancement of CQDs is associated the interaction between ionic conjugated system in SDBS and the π–π* and n–π* transition of CQDs needs further analysis. Moreover, laundry powder usually contains more than 20 components [1], and the residue of laundry powder may contain not only SDBS but also other ingredients. Thus we develop a quantified method to trace laundry powder by using a commercial laundry powder but not SDBS. 

A linear relationship between CQDs and laundry powder was observed. The enhanced fluorescence intensity of CQDs induced by laundry powder could be used to trace residues of laundry powder in water samples. The water pollutants in river water contains bacteria, viruses, parasites, fertilisers, pesticides, nitrates, phosphates, plastics, faecal waste and others [41]. Virus did not induce increased fluorescence intensity of the CQDs. And residue of laundry powder can be detected in river water, mineral water and tap water with the recovery rates more than 85%. This suggested that the prepared CQDs could be selectively detect the laundry powder in some real waters. However, whether high doses of other water pollutants affected on this assay need further analyze. Therefore, the prepared CQDs could be used as a fluorescent probe to detect residue of laundry powder in tap water, mineral water and tap water.

In summary, a synthesis method for CQDs was developed using waste bones from chicken. The prepared CQDs possess unique optical properties that may allow them to be used as a fluorescent probe to detect laundry detergents. This method could assist the low-cost and environmentally friendly reuse of waste biomass.

Reviewer 2 Report

Dear Editor

The manuscript ID:  molecules-1922964., entitled “A hydrothermal method to generate carbon quantum dots from waste bones
and their selective detection of laundry powder
” was carefully reviewed, and provided comments are listed as below. The manuscript reports the use of hydrothermal mediated carbon quantum dots from waste chicken bones for selective detection of laundry powder. The materials were characterized by using UV-vis and transmission electron microscope. The results are interesting and important, and there are good connections between the reported data and the discussion part. However, the manuscript needs further improvement and I recommend the publication of this manuscript after major revisions of the following comments.

1.      Figure 2 Authors should show distinguish between the two TEM images and label the caption correctly to reflect the distinctions made.

2.      Figure 3 needs improvement. The inset figures and caption are not readable

3.      Authors should explain the reason for the choice of the real samples used for the studies. Are this samples close to an industry where laundry powder is produced?

4.      I will suggest the addition of EDS analysis data for further confirmation the synthesized carbon quantum dots from chicken bone. This will show the percentage composition of C in the nanomaterial.

5.      Authors should add other characterization studies like Near Infra-red and photoluminescence.

Author Response

 Figure 2 Authors should show distinguish between the two TEM images and label the caption correctly to reflect the distinctions made.

Thank you. We modified this Figure

Figure 3 needs improvement. The inset figures and caption are not readable

Thank you. We modified this Figure

Authors should explain the reason for the choice of the real samples used for the studies. Are this samples close to an industry where laundry powder is produced?

Thank you for you question. We add the explain in the introduction parts.

The animal bone is an easily accessible waste during the meat processing activity. Slaughterhouses produced about 130 billion kg of animal bone residue globally every year [31]. In most developing countries, the waste bones are disposed of without regard to sound environmental management practices, making them harmful to humans and other terrestrial and aquatic life [32]. Thus, developing a method to generate CQDs from animal bone to detect the laundry powder is low-cost and environmentally friendly approach.

I will suggest the addition of EDS analysis data for further confirmation the synthesized carbon quantum dots from chicken bone. This will show the percentage composition of C in the nanomaterial.

Thank you for the comments. In the Figure 3c, d, e. We use XPS to analyze CQDs. The functional groups were identified and the percentage composition of C in the prepared CQDs is 75.34%. We modified the figure 3 to make it more readable.

 Authors should add other characterization studies like Near Infra-red and photoluminescence.

Thank you very much. In Figure 3b, we use Fourier transform infrared spectrometry (FT-IR) analysis of CQDs. FTIR is mainly used to measure light absorption of so-called mid-infrared light, light in the wavenumber range of 4,000 to 400 cm-1.

We modified the figure 3 to make it more readable.

For the photoluminescence, the data was moved to Figure 4a, and we modified this figure to make it more readable.

Round 2

Reviewer 1 Report

Although the authors made an effort to address my comments, I feel like they did not succeed.

First of all, the proof for selectivity still does not exist. To test two more chemicals is not enough. The word selective should not be used for this method description.

More importantly, the discussion and conclusion are unacceptable. This section should be improved in a scientifically satisfying way, with a thorough discussion of the presented results and their comparison with the existing literature.

Author Response

Comments and Suggestions for Authors

Although the authors made an effort to address my comments, I feel like they did not succeed.

First of all, the proof for selectivity still does not exist. To test two more chemicals is not enough. The word selective should not be used for this method description.

Thank you for your comments.

Firstly, As we wrote in the discussion parts “Considering that laundry powder contains more than 20 components [1] (main component SDBS and other ingredients) and other interferences in water samples, the influence on the fluorescens intensity was investigated. The results demonstrated that SDBS and laundry powder could significantly increase the fluorescence intensity of the CQDs, whereas other ingredients and water pollutants or interferences such as SDS, virus, and NP40 didn’t showed increased fluorescence intensity. Furthermore, real water samples were applied to testify the CQDs probe in detection of laundry powder with recovery rates more than 85%, which suggested that the prepared CQDs could successfully detect the laundry powder in some real water samples. “, we think that the prepared CQDs could detect the laundry powder even in river water.

Secondly, For the “selective” detection, we delete the word “selective” in the manuscript. We agree that it is very hard to completely understand that CQDs as prepared could selectively detect laundry powder, since the laundry powder is a mixture, and it is very hard to find the similar but different mixture.

More importantly, the discussion and conclusion are unacceptable. This section should be improved in a scientifically satisfying way, with a thorough discussion of the presented results and their comparison with the existing literature.

Thank you very much. We revised this part as following.

It has been widely reported that the CQDs can be produced from various kinds of biomass including plants and their derivatives such as corn powder [40] and green tea [41]; animals and their derivatives such as crab shell [42] and honey [43]; and municipal waste such as coal [44]. Herein, a one-step hydrothermal synthesis method was developed to generate CQDs using waste bone from chicken as raw material. Hydrothermal synthesis is a chemical reaction method in aqueous solution under the conditions of temperature of 100 - 1000 â—¦C, and the product has high purity, good dispersion, and easy particle size control [45]. The size of CQDs generated form crab shell by hydrothermal method is 4.0 ± 0.7 nm [42] and the average size of CQDs form lemon juice by hydrothermal method is 3.1 nm [27]. Consistently, the prepared CQDs had good water solubility and an almost monodisperse quasispherical structure with an average particle size of 3.2 ± 0.2 nm, which conforms to the ideals of green chemistry.

The optimal fluorescence excitation wavelength of the CQDs was 380 nm, and the corresponding emission peak was 465 nm. Therefore, emission of blue-green light was visible by the naked eyes under 380 nm UV light irradiation. The optical properties of prepared CQDs are quite similar with some published literatures [27,42,45,46]. For example, CQDs synthesized from organic waste exhibits a 450nm bule fluorescent light [47].  Besides the blue light with emission wavelengths of 450–475, the emission wavelengths of CQDs were observed in the range of 380–450, 476–495, 495–570, 570–590, 590–620, and 620–750 nm [45]. However, the PL mechanism of CQDs from biomass remains largely unknown. CQDs from the 3,4,9,10-perylenetetracarboxylic dianhydride with the sizes of 3.01±0.32 and 4.32±0.38 exhibit 400 and 480 nm fluorescence. Their PL mechanism is associated with the function groups including –COOH, –NH2, and C–N on the surface which induce fluorescence via the defect-induced luminescence mechanism of conjugated π structure [48]. Herein, the functional groups of prepared CQDs contain -OH, C=O, C=C and C-O, which correspond to the π–π* and n–π* transition [49]. Increased fluorescence intensity of the CQDs was observed after addition of SDBS but not SDS and NP40, which have quite similar structure as SDBS. Compared to SDS and NP40, SDBS contains an ionic conjugated system (Fig. 7b). Whether SDBS-induced fluorescence enhancement of CQDs is associated the interaction between ionic conjugated system in SDBS and the π–π* and n–π* transition of CQDs needs further analysis. Thus, this conjugated structure in CQDs is probably associated with its optical properties and PL mechanism.

Considering that laundry powder contains more than 20 components [1] (main component SDBS and other ingredients) and other interferences in water samples, the influence on the fluorescens intensity was investigated. The results demonstrated that SDBS and laundry powder could significantly increase the fluorescence intensity of the CQDs, whereas other ingredients and water pollutants or interferences such as SDS, virus, and NP40 didn’t showed increased fluorescence intensity. Furthermore, real water samples were applied to testify the CQDs probe in detection of laundry powder with recovery rates more than 85%, which suggested that the prepared CQDs could successfully detect the laundry powder in some real water samples. 

CQDs from biomass were widely used to detect the metal ions [50,51]. The CQDs from pear juice could selective detect Fe3+ [52]. On the other hand, non-metal ionic molecules such as Sudan I [53] and promethazine hydrochloride [54] can be detect by the CQDs from different biomass. Our work herein reported that CQDs such prepared could detect laundry powder and SDBS. Moreover, CQDs were reported to improve the photocatalysis to remove environmental pollutants [55]. Nitrogen/sulfur codoped carbon quantum dots were reported to detect sulfide and ferric ion which may be used to detect inorganic pollutants in water samples  [56]. CQDs have been combined with TiO2 was reported to disinfection of wastewater [57]. Thus, CQDs are promising nanomaterials for water pollution treatment [15].

In summary, a synthesis method for CQDs was developed using waste bones from chicken. The prepared CQDs possess unique optical properties that may allow them to be used as a fluorescent probe to detect laundry detergents. This method could assist the low-cost and environmentally friendly reuse of waste biomass.

Reviewer 2 Report

Dear Editor

The authors have worked on the manuscript, and the revised version is ok in its present form.

I recommend the publication of the manuscript.

Thanks

Fayemi

Author Response

Thank you for your revision for our work.